# Fluoride Exposure Induces Inhibition of Sodium/Iodide Symporter (NIS) Contributing to Impaired Iodine Absorption and Iodine Deficiency: Molecular Mechanisms of Inhibition and Implications for Public Health

**DOI:** 10.3390/ijerph16061086

**Published:** 2019-03-26

**Authors:** Declan Timothy Waugh

**Affiliations:** EnviroManagement Services, 11 Riverview, Doherty’s Rd, Bandon, Co. Cork P72 YF10, Ireland; declan@enviro.ie; Tel.: +353-23-884-1933

**Keywords:** fluoride, iodine, NIS, Na+, K+-ATPase, TNF-α, TGF-β1, IL-6 and IL-1β, IFN-γ, IGF-1, PI3K, megalin, prolactin

## Abstract

The sodium iodide symporter (NIS) is the plasma membrane glycoprotein that mediates active iodide transport in the thyroid and other tissues, such as the salivary, gastric mucosa, rectal mucosa, bronchial mucosa, placenta and mammary glands. In the thyroid, NIS mediates the uptake and accumulation of iodine and its activity is crucial for the development of the central nervous system and disease prevention. Since the discovery of NIS in 1996, research has further shown that NIS functionality and iodine transport is dependent on the activity of the sodium potassium activated adenosine 5′-triphosphatase pump (Na+, K+-ATPase). In this article, I review the molecular mechanisms by which F inhibits NIS expression and functionality which in turn contributes to impaired iodide absorption, diminished iodide-concentrating ability and iodine deficiency disorders. I discuss how NIS expression and activity is inhibited by thyroglobulin (Tg), tumour necrosis factor alpha (TNF-α), transforming growth factor beta 1 (TGF-β1), interleukin 6 (IL-6) and Interleukin 1 beta (IL-1β), interferon-γ (IFN-γ), insulin like growth factor 1 (IGF-1) and phosphoinositide 3-kinase (PI3K) and how fluoride upregulates expression and activity of these biomarkers. I further describe the crucial role of prolactin and megalin in regulation of NIS expression and iodine homeostasis and the effect of fluoride in down regulating prolactin and megalin expression. Among many other issues, I discuss the potential conflict between public health policies such as water fluoridation and its contribution to iodine deficiency, neurodevelopmental and pathological disorders. Further studies are warranted to examine these associations.

## 1. Introduction

Iodine is a vital micronutrient required at all stages of life; foetal life and early childhood being the most critical phases of requirement. Iodine, as its water-soluble iodide ion is the rate-limiting substrate for the synthesis of thyroid hormones (THs): thyroxin (T4) and tri-iodothyronine (T3), which are necessary for the control of cellular metabolism, growth and development of the body structures, neuronal function and development [1,2]. The T4 and T3, which are iodinated molecules of tyrosine, an amino acid that is synthesized in the body from phenylalanine, that is an essential amino acid, regulate oxidative enzymes and hence affect calorigenesis, thermoregulation, and intermediary metabolism. In addition to regulating carbohydrate metabolism these hormones also stimulate protein synthesis [3]. Iodine deficiency occurs when iodine intake falls below recommended levels and the thyroid gland is no longer able to synthesize sufficient amounts of thyroid hormone. The resulting hypothyroidism can occur at any stage of life, but the most devastating consequences of iodine deficiency take place during foetal development and childhood, with stillbirth, miscarriages, poor growth, and cognitive impairment. Iodine deficiency remains a major public health problem worldwide and the world’s greatest single cause of preventable brain damage [4].

Urinary excretion of iodine is the diagnostic indicator used to determine the nutritional state of iodine in the population, using the established references by WHO. Iodine deficiency is defined by the WHO as a population median urinary iodine concentration (UIC) that falls below 100 μg/L, while a median UIC of 50–99, 20–49 and <20 μg/L indicates mild, moderate, and severe iodine deficiency, respectively. In pregnancy the recommended threshold for UIC is 150 μg/L [5]. In addition to UIC, other measures of iodine status include thyroid volume, thyrotropin (TSH), triiodothyronine (T3), and thyroxine (T4). In 2004, it was estimated that of the two billion people around the world at risk of iodine deficiency, 20 percent live in Europe, Eastern and Western Europe being both affected [6]. In 2015, an estimated 12 countries have excessive iodine intake, 116 have adequate iodine nutrition and 25 remain iodine deficient [7,8].

While it is acknowledged that iodine deficiency increases the risk of fluoride (F) induced toxicity on thyroid function [9], it has also been reported that dietary iodine absorption and incorporation is reduced by F [10,11,12,13,14,15,16]. Indeed, in 2002, the Scientific Committee on Food, the main committee providing the European Commission with scientific advice on food safety, reported that dietary iodine absorption and incorporation is reduced by F in food and water [11]. Yang et al. observed that thyroid iodine uptake was markedly reduced in children when urinary F levels were approximately 2.0 mg/L. In this study, higher F exposure was also associated with dental fluorosis, higher serum TSH and lower IQ than age matched controls from a low F area [15]. Susheela et al. reported that elevated F uptake may cause iodine deficiency in fluorotic individuals, even when they reside in non-iodine deficient areas [12]. More recently, Sarkar and Pal suggested that F intoxication may contribute to iodine deficiency by inhibition of absorption of the iodine in humans as well as contributing to decreased retention of iodine through the interaction of F. [13]. A recent cross-sectional study conducted in Canada, which utilized weighted population-based data from Cycle 3 (2012–2013) of the Canadian Health Measure Survey (CHMS), found that UIC’s were lower in fluoridated than non-fluoridated communities. Moreover, iodine deficient individuals were found to have higher urinary F concentrations compared with the non-iodine deficient group [14]. This finding is supported by epidemiological data from China [17], which found that higher urinary F excretion was associated with significantly lower UIC in children. Further studies conducted in China, also found that excessive F exposure inhibited thyroid iodine uptake in children [10]. Similar effects were observed in a Russian study examining the effects of occupation exposure to F among subjects with signs of chronic fluorosis [18]. Consistent with these findings, a recent all Ireland study which measured iodine status in adolescent girls throughout the island of Ireland; which includes the Republic of Ireland (RoI) and Northern Ireland (NI), found that median UIC’s were significantly lower in adolescent girls in all participating locations in the RoI compared to NI. In addition, the percentage of adolescent girls with moderate to severe iodine deficiency (<50 µg/L) were 2–4-fold higher in the RoI compared to NI [19]. In examining this data, it is of fundamental importance to understand that approximately 84% of all households in the RoI have fluoridated water supplies [20], while drinking water is non-fluoridated in NI. Thus, the population in the RoI have higher exposure to F than NI. Therefore, this data validates the findings of the aforementioned studies suggesting that F exposure directly affects the bioavailability of iodine in humans.

It is important to be aware that chronic iodine deficiency increases the TSH concentration and produces a thyroid hormone pattern consistent with subclinical hypothyroidism [21]. Interestingly, in recent decades there has been a remarkably decline in UICs reported in the United States [22], Australia [23], New Zealand [24,25,26] and the RoI [27], leading one to postulate whether water fluoridation, which is widely practised in these countries, may be contributing to iodine deficient states and subclinical and overt hypothyroidism in the population. Indeed, recent studies conducted in England and Iran support this hypothesis [28,29]. Previous animal models of F-induced hypothyroidism have further demonstrated that excessive intake of F in drinking water and prenatal F intoxication induces hypothyroidism in offspring [30,31,32]. In another study, Ahmed et al. showed that the hypothyroid status during pregnancy and lactation produced inhibitory effects on monoamines, acetylcholinesterase (AchE) and ATPases and excitatory actions on γ-Aminobutyric acid (GABA) in different brain regions of the offspring. The authors reported that the thyroid gland of offspring of hypothyroid group exhibited histopathological changes as luminal obliteration of follicles, hyperplasia, fibroblastic proliferation and some degenerative changes [33]. Taken together, these findings suggest that F induced maternal hypothyroidism may cause a number of biochemical disturbances in different brain regions of offspring that may lead to a pathophysiological state.

Evidence from epidemiological studies also suggest that the combination of iodine deficiency and F exposure result in higher risk of developmental disorders, including impaired cognitive function in children [34,35,36]. Another experimental study found that excessive maternal exposure to F combined with a low iodine diet resulted in significantly increased oxidative stress in the brain of offspring compared to controls or animals exposed to F without iodine deficiency. Moreover, the authors of this study concluded that high F and low iodine intake resulted in significantly increased neurotoxic histopathological changes in the brain of offspring compared to high F alone, indicating that excessive F exposure and iodine deficiency interact synergistically to induce brain damage [37]. Consistent with these findings experimental research has also found that the combination of high F and low iodine has a greater negative effect on learning-memory of offspring rats than treatment with either low iodine or high F alone [38]. Further animal studies found that increased F intake combined with iodine deficiency can decrease serum thyroxine (T4) and triiodothyronine (T3) levels, as well as increase the F content and toxic effects of fluorosis in teeth and bones [39]. Experimental studies have also demonstrated that excess F intake causes DNA damage in thyroid cells comparable to that observed with iodine deficiency and that the effect was enhanced when F was given with a low iodine diet [40]. Another study with female mice and their lactating pups found that exposure to F resulted in a significant decrease in thyroid iodine content in pups and mothers along with hypertrophy of their thyroid glands. Notably, in this study, the authors reported that when F ion was eliminated from the mothers’ drinking water essentially complete recovery was observed in thyroid iodine content [31]. The potential of F to impair iodine stores has also observed in studies with cows, sheep and rodents resulting in significant reduced protein bound iodine in animals with chronic fluorosis compared to healthy controls from non-fluorotic areas [39,41,42,43].

As elucidated earlier, while studies among populations with different genetic variations in different geographic regions consistently show that F intake contributes to lower bioavailability of iodine, the molecular mechanisms by which this phenomenon occurs have not been fully defined. I recently revealed the molecular mechanisms by which F inhibits Na+, K+-ATPase activity and hypothesized that inhibition of Na+, K+-ATPase activity may play a crucial role in contributing to impaired iodine absorption and iodine deficient states. In this current study, I elucidate the role F plays in contributing to impaired iodine absorption focusing on the Na+/I- symporter (NIS). Here, I focus on the molecular mechanisms by which F inhibits NIS functionality and gene expression. These findings provide unprecedented insights into the molecular mechanisms by which F inhibits iodine uptake, transport and reabsorption, and provide a basis for minimising or preventing the risk of adverse health outcomes associated with iodine deficiency.

### 1.1. Dietary Sources of Iodine

Oceans are the world’s main repositories of iodine and very little of Earth’s iodine is actually found in the soil. Sea water contains 50 µg iodine/L. In the ocean, iodide is converted into elemental iodine, a volatile form that enters the environment as sea spray or volatile gases. Molecular iodine is also released from seaweed. After emission and due to photolytic dissociation iodine from the ocean is transferred to the atmosphere and returned to the land through rain and to a lesser extent snow [44,45,46,47,48,49,50,51]. Hence, it has been reported that coastal regions of the world are generally richer in iodine content than the soils further inland [52] and proximity to the sea, particularly coastal areas abundant with seaweed may contribute to dietary iodine intake through respiration [53,54].

The iodide content of foods and total diets differs depending on geochemical, soil, and cultural conditions. Notwithstanding the apparent contribution of marine atmospheric sources, the major natural sources of iodine are iodine saltwater fish and seafood because of their ability to concentrate iodine from seawater [52]. In certain countries such as Japan, the consumption of seaweed provides a significant source of dietary iodine [55]. Milk and dairy products are also a rich source of iodine, particularly in Ireland, the UK, Europe and North America where animal fodder is fortified with iodine and iodine-based disinfectants may be used in milking [56,57,58,59,60]. Other sources are eggs, freshwater fish, poultry and meat, fruits, legumes and vegetables. Generally common food sources provide 3–80 mg/serving [61,62]. The iodine content in drinking water can vary significantly depending on whether it is sourced from groundwater or natural surface waters, geology, aquifer characteristics and distance from sea. In certain geographic locations high iodine content in drinking water may also provide a major direct source of iodine intake, which may determine regional variations in iodine intake levels [63,64]. The iodine content in rivers can also vary depending on receiving effluent from urban areas [65]. Iodized salt provides another major source of iodine. Salt ionization was introduced in the USA, Australia and New Zealand in the 1920s and iodization policies have been increasingly used worldwide in recent decades to supplement dietary intake. The greatest access to salt iodization occurs in the World Health Organization (WHO) regions of the Western Pacific and the Americas, and the least access for those residing in the Eastern Mediterranean and Europe [4]. Since the 1950s the proportion of U.S. households which use only iodized salt has remained stable at 70–76% [66]. The intake of processed food containing iodized salt, calcium iodate, potassium iodate or cuprous iodide also increase the iodine intake. Non-food sources of iodine include iodine-containing medication, topical medicines, antiseptics, mineral dietary supplements, tablets or capsules of seaweed-based dietary supplements and kelp tablets as dietary supplements.

### 1.2. Metabolism of Iodine

Iodide is metabolized in the human body through a series of stages involving the hypothalamus, pituitary, thyroid gland and blood. The hypothalamus releases thyrotropin-releasing hormone (TRH) into circulation which stimulates thyrotrophs of the anterior pituitary to secrete TSH. regulates the secretion of thyroid-stimulating hormone (TSH) in the pituitary gland. In turn, TSH stimulates the thyroid follicular cells to release T4 and T3. When T4 is released into circulation, it can be converted to T3 through the process of de-iodination. TSH also stimulates the expression of thyroglobulin (Tg) and promotes the rapid internalization of Tg by thyrocytes. Altered regulation or defects in any of these steps can affect thyroid hormone synthesis and secretion [67]. Consumed iodine is absorbed as iodide in the intestines through the sodium iodide symporter (NIS) [68] and transferred into circulation. In contrast, the salivary glands and stomach take iodide from the bloodstream and release it into gastrointestinal tract for re-absorption through intestine [69].

Thyroid gland plays a central role in the metabolism of iodine. The gland comprises multiple follicles lined by follicular cells resting on a basement membrane. The follicles are filled by a clear viscous material called colloid. The colloid contains a glycoprotein called thyroglobulin. Iodine trapping is the first step in the metabolism of iodine [70]. Iodine transport by NIS from the bloodstream into thyrocytes through the basal surface is one of the most important events in thyroid hormone synthesis. In the thyroid, NIS concentrates iodide to a level that is 20 to 50 times that in plasma [71,72]. NIS also mediates the transport of iodine across cellular membranes and uptake of iodide into salivary, gastric mucosa, rectal mucosa, bronchial mucosa, placenta and mammary glands, through which iodide is delivered to the new-born for proper development [68,73,74,75,76,77]. Moreover, NIS regulates iodine availability in the thyroid, necessary for Tg iodination and T4 and T3 synthesis [78]. NIS functionality and iodine transport is dependent on the activity of the sodium potassium activated adenosine 5′-triphosphatase pump (Na+, K+-ATPase), or sodium pump [78]. Synthesis of T4, the primary form of TH released by the thyroid gland, consists of two sequential steps: iodination of selected tyrosines of Tg, and coupling of two doubly iodinated tyrosines within Tg to produce T4. Thus, TH synthesis relies on iodide availability [78]. To minimize the deleterious effects of iodide deficiency, iodinated Tg is stored extracellularly [78] and this process is mediated by megalin [79,80,81]. In the thyroid, megalin is expressed on the apical surface of thyrocytes, where it binds and internalizes Tg, after which Tg is transported across cells by transcytosis. Transcytosis is the major route by which Tg reaches the circulation. Thus, megalin plays an essential role in transcytosis and points to a major role of megalin in thyroid homeostasis with possible implications in thyroid diseases [79].

Because Tg is the molecular scaffold for TH synthesis and iodide storage, it is important to note that Tg molecules internalized by megalin are not reduced or degraded in the lysosomal pathway which otherwise lead to the formation of T4 and T3 [80]. This extracellularly storage serves as the body’s primary reservoir for iodide storage [78]. Thus, megalin plays a crucial role in regulating iodine stores and the thyroid exchangeable hormonal iodine pool. During periods of low iodine intake or deficiency TSH secretion is augmented which stimulates Tg reabsorption by thyrocytes, Tg degradation and the release of iodine for the synthesis of TH. [82,83]. It is also known that albumin is iodinated in the thyroid gland and when albumin is elevated in blood a high protein bound iodine value can be recorded [84].

The thyroid is estimated to use 60–80 μg of iodide daily to produce its customary output of thyroid hormones [85]. About a quarter of the daily requirement of iodine is acquired from recycling endogenous iodide and the rest is acquired from the diet [86]. In healthy adults, the absorption of iodide is greater than 90% [87]. In conditions of adequate iodine supply, no more than 10% of absorbed iodine is taken up by the thyroid. In chronic iodine deficiency, this fraction can exceed 80% [88,89,90]. In most individuals, if iodine intake falls below approximately 100 µg/d, TSH secretion is augmented which increases plasma inorganic iodide clearance by the thyroid through stimulation of NIS expression. As a greater fraction of circulating iodine is cleared by the thyroid, there is a progressive reduction in renal iodide excretion. [91]. Chronic iodine deficiency increases the TSH concentration and produces a thyroid hormone pattern consistent with subclinical hypothyroidism [21]. Iodine deficient individuals typically demonstrate a variable elevated TSH, with a low-normal range of serum T4, and a normal or high-normal T3 [91].

In pregnancy the iodine requirement is increased due to an increase in maternal T4 production to maintain maternal euthyroidism and transfer thyroid hormone to the foetus early in the first trimester, before the foetal thyroid is functioning; iodine transfer to the foetus, particularly in later gestation; and an increase in renal iodine clearance [92]. Foetal thyroid activity depends entirely on the availability of iodine transferred from maternal circulation [93]. During lactation, the mammary gland concentrates iodine and secretes it into breast milk to provide for the new-born, thereby supplementing the iodine pool in infants for the synthesize TH [94]. The infant also needs a supply of iodine for normal thyroid activity, vital for brain development in the first 2 years of life. The supply of iodine to the neonate and infant comes exclusively from breast or formula milk in the first 6 months of life and from milk/formula and complementary foods thereafter [94]. Thus, the iodine requirement of a woman who is fully breastfeeding her infant is likely to be higher than that during pregnancy [91]. The lactating breast can concentrate iodide to a similar degree as that seen in the thyroid, producing milk with an iodine concentration of 20–700 μg/L [95].

## 2. Molecular Mechanisms of Fluoride Inhibition of Iodine Homeostasis

The molecular mechanisms of F action underlying disturbed iodine homeostasis are complex and require a detailed knowledge of regulation of biological processes. These effects derive from two interlinked mechanisms. As I previously elucidated, F acts to inhibit Na+, K+-ATPase activity, which is essential for regulating NIS functionality [96,97]. NIS expression and functionality in turn is required for the efficient absorption of iodide in the intestines, and the transportation and uptake of iodide into the thyroid gland. As previously noted, NIS also mediates the transport of iodine across cellular membranes and uptake of iodine into salivary, gastric mucosa, rectal mucosa, bronchial mucosa, placenta and mammary glands. However, current evidence also suggests that F acts to inhibit NIS mRNA expression through several biological mechanisms as outlined below.

### 2.1. Molecular Mechanisms of F Inhibition of Na+/K+-ATPase Activity

As a first step to elucidate the basis for the decrease in iodine bioavailability observed in countries with fluoridation programmes, it is of fundamental importance to understand that NIS functionality requires Na+, K+-ATPase activity. As previously mentioned, the molecular mechanisms of F inhibition of Na+, K+-ATPase activity have recently been comprehensively defined [98]. In summary it has been shown that activation of PKC, cAMP, cGMP, NO, Pi, PLA2, AA, PGE2, dopamine, glucose and PTH and the formation of AGEs inhibit Na+, K+-ATPase activity and that F acts to upregulate the formation of these biomarkers. In addition, it has been elucidated that F inhibition of CT contributes to impaired Na+, K+-ATPase functionality [96].

### 2.2. Deciphering the Molecular Mechanisms of F Inhibition of NIS Expression and Activity

Several cytokines have been identified to inhibit NIS mRNA expression and activity including tumour necrosis factor alpha (TNF-α), transforming growth factor beta 1 (TGF-β1) [97,98,99,100,101,102], IL-6 and IL-1β [103], and interferon-γ (IFN-γ) [104]. It has also been documented that the hormone, insulin like growth factor 1 (IGF-1) [105], and enzyme phosphoinositide 3-kinase (PI3K), inhibit NIS mRNA expression in thyroid cells [106]. Several studies have shown that the protein thyroglobulin (Tg), decreases NIS gene [107,108,109,110,111,112,113] and vascular endothelial growth factor (VEGF) gene expression [109], thereby reducing iodine uptake. Tg also supresses gene expression of TSH receptor (TSHR) [108,112,113] and thyroid transcription factor-1 (TTF-1) [108,113]. Previous studies have also shown that TTF-1 upregulates NIS gene expression [114,115,116]. Moreover, TSHR is regulated only by TTF-1 [108], thus, loss of TTF-1 can also result in impaired TSHR functionality. Furthermore, loss of TSHR abundance or function can result in partial TSH resistance, increased serum TSH levels and hypothyroidism [117]. Inactivation of TSHR receptor is also associated with premature birth and low birth weight babies [118]. Conversely, prolactin (PRL) has been found to elevate NIS mRNA and protein levels [119,120,121]. As there are no studies as yet specifically addressing F inhibition of NIS activity, evidence was sought from published literature including human, animal and in vitro studies to examine how F interacted with each of the aforementioned biological pathways identified as inhibitors of NIS gene expression or activity. A schematic representation of the role of NIS in iodine transport and the molecular mechanisms of NIS inhibition is provided in Figure 1. In the following section, I summarise the studies documenting the effect of F on TNF-α, TGF-β1) IL-6 and IL-1β, IFN-γ, IGF-1, PI3K and Tg.

Though the underlying molecular mechanisms remain elusive, previous studies have shown that F activates IGF1 signalling [122], resulting in significantly increased serum IGF-1 levels [123]. F has also been found to enhance the mitogenic activities of IGF-1 action on bone cells [124,125]. However, it is important to note that prostaglandin E2 (PGE2) and TNF-α have been shown to induce IGF-1 production [126,127,128,129,130]. PGE2 acts as a positive stimulus for IGF-I synthesis through a cyclic AMP/PKA pathway and TNF-α stimulated IGF-1 synthesis via the mitogen-activated protein (MAP) kinase pathway [130]. Of fundamental importance, in vitro experiments using human blood cells have consistently demonstrated that F in micromolar concentrations of 1–10 µM significantly increased the synthesis of cAMP and the synthesis of PGE2 in a dose dependent manner [131,132,133]. Most recently, Gutowska et al. demonstrated that F at 3 µM increased PGE2 synthesis by 42% in macrophages compared to negative controls [133]. Several in vitro, in vivo and human studies have also demonstrated that F induces upregulation of IL-6 and IL-1β mRNA expression resulting in enhanced activity [134,135,136,137,138]. In addition, F exposure has been found to upregulate TNF-α expression in human and animal studies [134,135,136,137,138,139], and it has also been demonstrated in vivo and in vitro that F induces upregulation of PI3K mRNA and protein levels [140,141,142,143,144].

Studies have also shown that F significantly increases IFN-γ mRNA expression and serum levels [145]. Notably, Lv et al. observed that F induced IFNγ signalling activates osteoclasts and aggravates oestrogen deficiency inducing osteoporosis [145]. It has also been reported that TGF-β1 plays an important role in fluorosis and increased levels of TGF-β1 have been suggested as an important marker in the evaluation of the pathological action of F in bone tissue [146,147]. In vivo and in vitro experimental studies of fluorosis have shown that F upregulates TGF-β1 protein and mRNA expression in bone cells [146,148,149,150]. Importantly, Calcitonin (CT), a hormone that is secreted by parafollicular cells of the thyroid gland has been found to be a potent stimulator of TGF-β1 protein synthesis as well as TGF-β1 mRNA expression. [151]. A large body of evidence from epidemiological studies has demonstrated that F is a potent inducer of CT expression in humans [152,153,154,155,156,157]. Of fundamental importance is the research by Chen and associates in providing a biological dose-exposure response relationship for F exposure on CT expression at relatively low intakes. Notably, in this study, it was demonstrated that differential expression of CT occurs when urinary F levels exceeded 0.38 mg/L [155].

As previously noted, PRL has been found elevate NIS mRNA and protein levels. It is important to note therefore that in vivo and in vitro studies with animals [158] as well as human studies [159,160] have shown that F inhibits PRL secretion. Moreover, the inhibitory effect was found to be significantly more prevalent in females than males [160]. Notably, among adults with dental fluorosis Murugan and Subramanian observed that in addition to an abrupt reduction in serum concentrations of PLC, subjects with dental fluorosis also had significantly lower levels of T3 and T4 and significantly higher levels of TSH compared to subjects residing in the same community without dental fluorosis [159]. In another study, Ortiz-Pérez et al. found that serum PRL levels were significantly lower in adult males occupationally exposed to F with a mean urinary F level of 3.2 mg/L (range 2.9–3.4 mg/L) compared to males residing in a low F community where the mean urinary F level was 1.6 mg/L (range 1.3–1.9 mg/L) [160]. While F has been found to inhibit PRL, the molecular mechanisms behind this of inhibition remain unclear. However, it is important to note that CT is a PRL-inhibiting hormone [151,161,162,163,164]. As previously discussed, F has been found to be a potent inducer of CT expression in humans [152,153,154,155,156,157]. These findings suggest a plausible mechanism whereby F exposure inhibits NIS expression through F stimulation of CT which acts to inhibit PRL secretion.

As previously elucidated, Tg has been found to play a crucial role in downregulating NIS expression. There is robust evidence from epidemiological studies conducted in different geographic regions to show that F exposure increases TSH levels in humans [12,15,18,29,35,165,166,167,168,169,170,171,172,173,174,175,176,177,178]. Since TSH stimulates thyroid follicular cells to produce Tg [179,180]; serum Tg concentrations are typically higher in individuals with a raised TSH or primary hypothyroidism [181]. TSH dependant elevation of serum Tg has also been observed in subclinical hypothyroidism in patients with chronic kidney disease [182]. Interestingly, evidence suggests that serum Tg levels are higher in females than males [183,184,185], and increase with increasing levels of TSH and decreasing urinary iodine concentration [185]. However, it is also important to note that NIS expression is inhibited by oestradiol, the major female sex hormone [186]. This may also explain why thyroid diseases predominantly affect women; their incidence is 5–20 times higher in women than in men [187]. Of note, oestrogen, the primary female sex hormone suppresses IFNγ [145]. In menopause oestrogen levels decrease, as oestrogen decrease IFNγ expression increases, which elucidates why the incidence of hypothyroidism increases significantly in postmenopausal women [188,189,190,191]. However, it is important to note that serum F levels have also been shown to increase significantly in women post menopause [192,193,194]. Husdan et al. reported that the rate of change of serum ionic F concentration with age for women was observed to be twice that for men over 45 years of age. It was suggested that the greater rate of increase noted in females was probably due to the enhanced release of F from bone observed after the menopause [192]. These results indicate that elevation of plasma F levels post menopause may be involved in elevating TSH as well as Tg and contributing to the increased NIS inhibition and increased prevalence of hypothyroidism in postmenopausal women.

It is important to note, that Tg gene expression is regulated by insulin and insulin like growth factor 1 (IGF-1) [195]. Moreover, the presence of insulin is necessary for TSH inhibition of TTF-1 gene expression [196]. Accumulating evidence has shown that osteocalcin, which is specifically expressed in osteoblasts and secreted into the circulation, regulates insulin secretion by stimulating insulin expression in pancreas [197,198,199,200,201,202,203], as well as by indirectly stimulating insulin through increasing the secretion of glucagon-like peptide-1 (GLP-1), an incretin released by intestinal endocrine cells [204,205]. Importantly, several epidemiological studies have also confirmed that F exposure significantly increases OC expression in humans [154,155,206,207,208,209,210]. A significant effect of F on OC expression has been found to occur in adults at serum F levels of 1.0 μM [154] or when urinary F levels exceed 0.62 mg/L in adolescent children [155]. Consistent with this data, several in-vivo studies have reported that chronic F exposure enhances plasma insulin levels and promotes insulin resistance in rodents [211,212,213,214,215]. A similar association has also been found in humans chronically exposed to F [216]. As previously noted, studies have also shown that F activates IGF1 signalling and enhances the mitogenic activities of IGF-1 action on bone cells [122,123,124,125].

Finally, it is important to mention one other protein, megalin. As previously elucidated megalin regulates iodine stores and the thyroid exchangeable hormonal iodine pool in humans by regulating extracellularly storage of iodine in Tg, which serves as the body’s primary reservoir for iodide storage. Furthermore, Na+, K+-ATPase activity is required for NIS to transport iodide into Tg for iodide accumulation and thyroid hormone synthesis in the thyroid gland. As previously noted, Na+, K+-ATPase is inhibited by F [96]. Thus, inhibition of megalin or Na+, K+-ATPase can lead to depletion of iodine stores and iodine deficiency. While there are no studies in literature examining the relationship between F exposure in humans and megalin expression, recently it has been found in animal models that chronic F exposure inhibits megalin expression [217]. It has been observed in animal studies that a decrease in megalin expression leads to a decrease in proximal tubule albumin (ALB) reabsorption [218], leading to enhanced urinary excretion. Further studies have found that megalin expression is inhibited by elevated glucose levels [219,220]. Importantly, a large number of human and animal studies have demonstrated that F exposure can induce hyperglycaemia [218,221,222,223,224,225,226,227,228,229,230,231,232,233,234]. Consistent with this, the U.S. National Research Council (NRC) reported that the conclusions from available studies is that sufficient F exposure appears to bring about increases in blood glucose or impaired glucose tolerance in some individuals and the increase the severity of some types of diabetes [9]. Therefore, evidence suggests that the ability to F to promote hyperglycaemia may lead to inhibition of megalin expression. Further studies are required confirm this association in humans.

Moreover, in addition to Tg, it is also known that ALB is iodinated in the thyroid gland and when ALB is elevated in blood a high protein bound iodine value can be recorded [84] and in cases of hypoalbuminemia (low levels of ALB in blood) serum iodine levels can be reduced [235,236]. These observations provide evidence of the role of ALB in modulating the iodine pool in humans. In this regard, it is long known that one of the clinical symptoms of F intoxication is inflammation of the kidneys, resulting in higher ALB in urine [237]. As previously described, inhibition of megalin expression can also lead to enhanced urinary excretion by impairing proximal tubule ALB reabsorption, which further leads to lower ALB in systemic circulation. Consistent with these findings, a recent cross-sectional study reported an association between F exposure and increased urinary excretion of ALB in Mexican adults (mean age 46 years). In this study, the median urinary F concentration was 2 mg/L with approximately 59% of urine samples having a F concentration ≥ 1.6 mg/L. [238]. Similarly, Kumar and Harper previously reported that evidence of renal damage, increased urinary ALB and vascular calcification were associated with skeletal fluorosis [239].

Human studies have also shown that circulating ALB levels of individuals with skeletal fluorosis as well as the concentration of salivary albumin in children with dental fluorosis are significantly lower than in healthy controls without fluorosis [240,241,242]. Studies have also shown that high blood glucose levels significantly reduce serum ALB levels [243], which is consistent with the previous observations that megalin expression is inhibited by glucose [219,220]. Also, as previously noted, the salivary glands play a role in recycling iodine for uptake in the gastrointestinal tract [69]. Thus, lower salivary ALB increased urinary excretion of ALB and lower blood levels of ALB in systemic circulation can contribute to depletion of iodine stores in the body and iodine deficiency. Taken together, this data may explain the findings of Singla and Shashi who observed increased urinary loss of iodine in patients with thyroid dysfunction residing in an endemic fluorosis area [13]. From a pharmacokinetic point of view, it has further been demonstrated that megalin excretion is enhanced in the urine of patients with moderate increase in the level of urine ALB [244]. This suggests that the binding of megalin to ALB and increased excretion of ALB which occurs during F intoxication, may further contribute to elimination of megalin with implications for homeostatic regulation of iodine. It is essential to also clarify that NIS is localized at the opposite apical (Ap) surface of the proximal and cortical collecting tubes in the kidney and participates in iodine reabsorption [245]. Accordingly, loss of NIS or inhibition of Na+, K+-ATPase which is required for NIS functionality, can result in impaired reabsorption of iodine, which can in turn contribute to iodine depletion. Interestingly, in hypothyroidism the activity of Na(+)-K(+)-ATPase is reduced initially in the proximal tubules and later in almost all segments of the nephron [246]. As previously described, iodine deficiency leads to increased TSH secretion which can result in hypothyroidism. It is important to mention that while, results from several animal and in vitro studies suggest that TSH directly stimulates NIS mRNA expression and protein levels in the thyroid gland [247,248,249,250], evidence from human studies have demonstrated that serum TSH levels have not been found to be associated with NIS expression [100,251,252].

## 3. Discussion

In this work, the key mechanisms by which F inhibits iodine absorption thereby contributing to iodine deficiency within the population have been presented. Evidence demonstrates that F impairs iodine absorption and transport by two key mechanisms; inhibition of NIS gene expression and inhibition of Na+, K+-ATPase activity. NIS mediates iodine uptake in the gastrointestinal tract as well as uptake of iodine into the thyroid, salivary, gastric mucosa and mammary glands. In relation to NIS functionality in iodine absorption and transport the crucial role of Na+, K+-ATPase activity has been elucidated. It is also known that F inhibits Na+, K+-ATPase activity and the molecular mechanism by which this occurs have recently been described [96].

In this study, evidence has been presented demonstrating NIS gene expression is activated by PRL and that F inhibits PRL expression. Importantly, it has also been elucidated that the inhibitory effect is significantly more prevalent in females than males. It has further been shown that CT is a PRL-inhibiting hormone and that F is a potent inducer of CT expression. Furthermore, evidence has been presented that TNF-α, TGF-β1, IL-6 and IL-1β, IFN-γ, IGF-1, PI3K and Tg inhibit NIS expression and that F upregulates TNF-α, TGF-β1, IL-6 and IL-1β, IFN-γ, IGF-1, PI3K and Tg expression and activity.

In addition to providing causal mechanisms by which F impairs iodine metabolism an important consideration in demonstrating a causal association between F intake and impaired iodine deficiency is evidence from epidemiological studies. In this regard, I have previously elucidated that evidence from several epidemiological studies undertaken in different geographic locations among different ethnic populations consistency shown an association between F exposure and impaired iodine bioavailability. Furthermore, evidence for animal studies support this observation. In addition, I have previously elucidated how the mean UICs have declined dramatically in recent decades in countries with advanced water fluoridation programmes.

It is important to note that the most obvious and the earliest clinical manifestation of F toxicity is dental fluorosis, which can develop in children, due to the intake of high levels of F during the period of tooth development, and clinically, it is the marker of F toxicity in the first six years of life [253]. The increasing population exposure to F is clearly evident in the rising burden of dental fluorosis as reported in the US and other countries with water fluoridation in recent decades. Revealing the dramatic decline in UICs observed in these countries in recent decades parallels the dramatic increase in the prevalence of dental fluorosis. For example, in the US, the prevalence of dental fluorosis was 9% among children born in the period 1961–1970, compared to 22.6% among children born between 1984–1985 and 41% among all US children born between 1984–1985 [254]. In 2004, Marshall et al. reported that the prevalence of fluorosis in permanent teeth in areas with fluoridated water in the USA increased from about 10–15% in the 1940s to as high as 70% in recent studies [255]. Similarly, among children aged 15 years the prevalence of dental fluorosis increased seven-fold in fluoridated communities in the RoI between 1984 and 2002 [256]. In this regard, it is important to identify that a comprehensive review of the available literature on disorders induced by iodine deficiency previously reported that iodine deficiency among the paediatric population may result in significantly lower iodine stores in the thyroid and significantly reduce the thyroid exchangeable hormonal iodine pool in adulthood [257].

In this study, I have also discussed the crucial role of megalin in regulating iodine stores and presented evidence to show that F inhibits megalin expression as well as elucidated the molecular mechanisms by which this occurs. I have further elucidated how iodine is also carried in the blood stream bound to ALB and that high serum ALB correlates with high protein bound iodine levels. I have described how the literature shows that F exposure results in loss of ALB in urine contributing to lower salivary and serum ALB levels. Additionally, I have discussed how chronic F exposure has been found to result in lowering of protein bound iodine levels, indicating that F intoxication is contributing to iodine deficiency by lowering iodine stores. I have also elucidated that evidence from animal studies show that increased F intake combined with iodine deficiency significantly enhances the toxic effects of F on teeth, bones, thyroid and brain. Taken together, these data may explain the dramatic increase in dental fluorosis observed in fluoridated communities in recent decades. Moreover, the evidence presented in this study offers a rational for the remarkably decline in UIC among females of reproductive age observed in countries with extensive water fluoridation programmes in recent decades. As mentioned previously, iodine deficiency in infancy can also lead to impaired iodine stores in adulthood.

As previously described, it is well recognized that impaired cognitive development in offspring is associated with maternal iodine deficiency [4]. In addition, a growing body of evidence exists that shows an association between iodine deficiency/hypothyroidism, attention deficit hyperactivity disorder (ADHD) [258,259,260,261,262] and autism spectrum disorders (ASD) [263,264,265,266,267,268]. As elucidated in this study, given the diverse role of F in inhibiting NIS expression and functionality, including the crucial role of F in inhibiting Na+ K+-ATPase activity, as recently reported [96], it is expected that maternal F exposure will increase the risk of neurodevelopmental disorders in offspring. Taken together, this suggests that F intake should be minimized in infancy, and especially among females prior to and during pregnancy to ensure adequate iodine stores are available for the developing foetus.

According to the WHO, in iodine-deficient communities, between 10 and 15 intelligence quotients (IQ) points may be lost when compared to similar but non-iodine-deficient populations [269]. Revealing, Lynn and Vanhanen examined differences in the average IQ of children for over 80 nations internationally for which data on the average national IQ were available. According to the authors, the median national IQ for the RoI was significantly below that of other European nations including the UK, The Netherlands, Austria, Germany, Sweden, Estonia, Belgium, France, Switzerland, Czech Rep, Denmark, Italy, Finland, Spain, Slovenia, Hungary and Ukraine [270]. Interestingly, the authors reported that between 1988 and 1993 the mean IQ among children aged 6–12 years in the RoI declined by 4 IQ points from 97 to 93 [270]. Furthermore, in 2012, the mean IQ of children aged 6 years in the RoI was reported to be 92. It is important to note that the drop in IQ follows the trend of the reduction in UIC among females of reproductive age [23] and the increasing prevalence of dental fluorosis among children born in RoI in recent decades [256] Furthermore, the RoI is the only country in Europe with mandatory artificial fluoridation of drinking water. As previously elucidated, the combination of low iodine status and high F intake has been found in animal studies to result in enhanced toxic effects of fluorosis on teeth [43] and brains [41]. The trend of declining IQ in children with increasing prevalence of dental fluorosis observed in the RoI is consistent with the findings of several cross-sectional studies conducted in China and India which documented an association between loss of IQ and dental fluorosis in children [271,272,273,274,275,276,277]. Taken together, these results suggest that water fluoridation and F intake has contributed to a significant loss of IQ in children in the RoI. This observation is consistent with the recent findings from two well designed longitudinal studies which found that prenatal exposure to F was associated with cognitive impairment and increased risk of ADHD in offspring [278,279].

## 4. Additional Perspectives

In addition to the biological actions of iodine in regulating thyroid hormones, iodine has many additional functions, including anti-inflammatory [280,281,282,283], antioxidant [284,285,286,287] and anti-microbial defence [288,289,290,291,292,293,294]. Iodine deficiency has been reported to be associated with increased cancer risk [295] including; breast [286,296,297,298,299], thyroid [300,301,302,303,304,305,306], and prostate cancer [307,308]. Furthermore, iodine has been found to inhibit the carcinogenic process in breast and prostate cancer cell lines [309,310,311,312]. These findings are consistent with results showing that incidence of prostate, endometrium, ovary and breast cancer is lower in populations with high iodine intake [245,299]. Notably, the RoI has the highest cancer incident in Europe and third highest in the world next to New Zealand and Australia [313,314]. The majority of the population in all three countries are also provided with artificially fluoridated drinking water.

The current study has elucidated the role of F in inhibiting PRL expression. It is important to note, that in addition the role of PRL in upregulating NIS expression, PRL is known to have more than 300 biological functions including the stimulation of neurogenesis, modulation of stress responses, calcium transport, immune system regulation and reduction of anxiety, among others [315,316,317,318,319]. Recently, it has been found that PRL increases synaptogenesis and neuronal plasticity, enhances neurogenesis, cell proliferation and neuroprotection, improves learning and spatial memory and acts as a neuronal protector against excitotoxicity-effects [320,321,322,323,324,325,326,327,328,329,330]. Thus, it seems reasonable to suggest that suppression of PRL may be deleterious to brain activity. However, PRL is also essential for promoting milk production after birth [331]. Thus, inhibition of PRL can lead to problems with breastfeeding. Low PRL is associated with increased risk of type 2 diabetes (TSDM), particularly for females [332,333]. Globally, the number of people with diabetes mellitus has quadrupled in the past three decades, and diabetes mellitus is the ninth major cause of death. About 1 in 11 adults worldwide now have diabetes mellitus, 90% of whom have T2DM [334]. In males, low PRL has been found to be related to sexual dysfunction, metabolic syndrome, anxiety and depressive symptoms and lower general health. [335,336]. In line with this finding, in a longitudinal evaluation of the same subjects, it was recently reported that low PRL independently predicted incident major cardiovascular events [337].

## 5. Conclusions

In summary, diverse lines of evidence demonstrate that F inhibits NIS expression and functionality thereby contributing to impaired iodide absorption, diminished iodide-concentrating ability and iodine deficiency disorders. Taken together, the findings of this study provide unprecedented insights into the molecular mechanisms by which F inhibits iodine uptake, transport and reabsorption, and provide a basis for minimising or preventing the risk of adverse health outcomes associated with iodine deficiency. The findings of this study further suggest that there are windows of susceptibility over the life course where chronic F exposure in infancy may influence long term iodine status and health outcomes in adulthood. Moreover, iodine deficiency in pregnancy can have profound implications on the health of offspring including risk of miscarriage, stillbirth, reduced IQ and greater risk of ADHD and ASD. In addition, loss of iodine bioavailability leads to lower antioxidant capacity, reduced anti-inflammatory capacity, impaired immunity and increased risk of cancer. Therefore, actions that prevent the risk of adverse health outcomes are essential to improve the health of the world’s population and to reduce health inequities. Based on the findings of this study, evidence strongly suggests that F ingestion contributes to pathological states by impairing iodine absorption and diminishing iodine concentrating ability. Whether or not public health practitioners and epidemiologists are ready to take up the implied challenge of these findings is, inevitably, related to the extent to which they are willing to continue to disavow the emerging evidence that F intake is associated with negative health outcomes for the sake of continuing a policy of fluoridation of drinking water that inadvertently may be contributing to the pathogenesis of neurodevelopmental disorders, impaired immune responses, inflammatory diseases and cancer.

## Figures and Tables

**Figure 1 ijerph-16-01086-f001:**
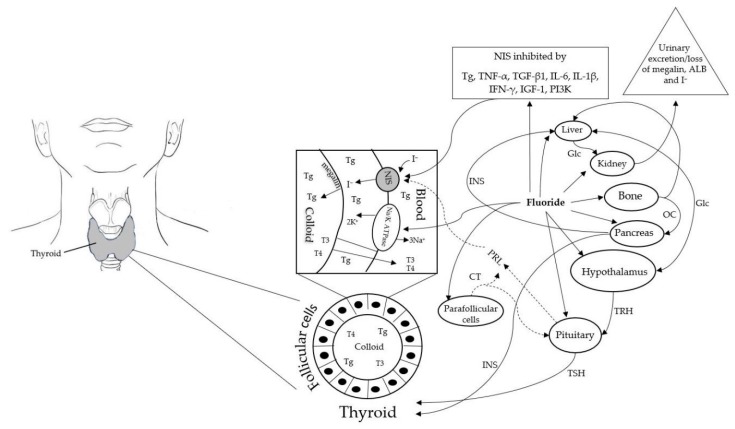
Schematic representation of the role of NIS and Na+, K+-ATPase in normal thyroid follicular cells showing the key aspects of iodine transport, thyroid hormone synthesis and molecular mechanisms of inhibition of NIS by fluoride. Abbreviations: NIS: Sodium/Iodide Symporter; ALD: Albumin; CT: Calcitonin; Glc: Glucose; I-: Iodide; INS: Insulin; IGF-1; Insulin like growth factor 1; IL-6: Interleukin 6; IL-1β: Interleukin 1 beta; IFN-γ: Interferon-γ; OC: Osteocalcin; PRL: Prolactin; PI3L: Phosphoinositide 3-kinase; Tg: Thyroglobulin; THR: Thyrotropin-releasing hormone; TSH: Thyroid-stimulating hormone; T4: Thyroxin; T3: Tri-iodothyronine.

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
