# Peer review of "Fluoride Exposure Induces Inhibition of Sodium/Iodide Symporter (NIS) Contributing to Impaired Iodine Absorption and Iodine Deficiency: Molecular Mechanisms of Inhibition and Implications for Public Health"

_ijerph, 2019, doi:10.3390/ijerph16061086_

Round 1

Reviewer 1 Report

The interesting manuscript "Fluoride Exposure Induces Inhibition of Sodium/Iodide Symporter (NIS) contributing to impaired Iodine Absorption and Iodine Deficiency: Molecular Mechanisms of Inhibition and Implications for Public Health" reviews the relationship between fluoride and thyroid iodide uptake. The author has found data in literature suggesting that fluoride may affect iodine availability for thyrocytes by interference with both Na+/K+ ATPase and NIS expression. Moreover, data also suggest that excessive water fluoridation could affect thyroid function, thus predisposing to neurodevelopmental disturbances. A figure or table regarding the mechanisms by which fluoride regulates NIS could be helpful to summarize all the data described. The subject is well within the scope of the journal. Some text-editing is necessary. 

Comments:

Introduction

First paragraph, lines 36-37: In the sentence “The T4 and T3, which are iodinated molecules of the essential amino acid tyrosine…”, tyrosine is a non-essential amino acid. In the same sentence “The T4 and T3, which are iodinated molecules of the essential amino acid tyrosine, regulate cellular oxidation…”, what does “cellular oxidation” mean?

First paragraph, line 38: “These hormones are necessary for protein synthesis”. Thyroid hormones stimulate the synthesis of some proteins, but this sentence seems to be very generic and should be rewritten.

Page 3, 132: “update” might be changed to “uptake”.

Page 4, Metabolism of Iodine, second paragraph, line 177: The sentence “The colloid is a gycoprotein called thyroglobulin” should be changed to “The colloid contains a gycoprotein called thyroglobulin”.

Same paragraph, lines 183-184: The sentence “Moreover, NIS regulates the iodination of Tg leading to T4 and T3 synthesis” might be changed to “Moreover, NIS regulates iodine availability in the thyroid, necessary for Tg iodination and T4 and T3 synthesis”.

Same paragraph, lines 189-190: “To minimize the deleterious effects of iodide deficiency, iodinated Tg stored extracellularly [79] and this process is mediated by megalin [80-82].” Megalin is important for Tg transcytosis. Is it involved in Tg storage too?

Page 5, second paragraph: “During lactation, the mammary gland concentrates iodine and secretes it into breast milk to provide for the new-born, until it can synthesize its own TH [96].” Thyroid hormones are synthetized by the fetus early in gestation. The sentence does not make sense.

Page 7, last paragraph, line 342: The sentence “…NIS uptake of iodine into Tg.” does not make sense and should be rewritten.

Page 10, first paragraph, line 455-456: The sentence “Taken together, this suggests that F intake be minimized in infancy…” might be changed to “Taken together, this suggests that F intake should be minimized in infancy…”

Page 11, conclusions, line 513: “…iodine update…” should be changed to “…iodide uptake…”.

Author Response

I would like to thank the reviewer for careful and thorough reading of this manuscript and for the thoughtful comments and constructive suggestions, which help to improve the quality of this manuscript.

General Comments. The interesting manuscript "Fluoride Exposure Induces Inhibition of Sodium/Iodide Symporter (NIS) contributing to impaired Iodine Absorption and Iodine Deficiency: Molecular Mechanisms of Inhibition and Implications for Public Health" reviews the relationship between fluoride and thyroid iodide uptake. The author has found data in literature suggesting that fluoride may affect iodine availability for thyrocytes by interference with both Na+/K+ ATPase and NIS expression. Moreover, data also suggest that excessive water fluoridation could affect thyroid function, thus predisposing to neurodevelopmental disturbances. A figure or table regarding the mechanisms by which fluoride regulates NIS could be helpful to summarize all the data described. The subject is well within the scope of the journal. Some text-editing is necessary.

Reply:

I appreciate the positive feedback from the reviewer. I have included Figure 1, a schematic illustration of the role of NIS and Na+, K+-ATPase in normal thyroid follicular cells showing the key aspects of  iodine transport, thyroid hormone synthesis and molecular mechanisms of inhibition of NIS. I have amended/edited text, as suggested.

Minor comments:

1)      Introduction. First paragraph, lines 36-37: In the sentence “The T4 and T3, which are iodinated molecules of the essential amino acid tyrosine…”, tyrosine is a non-essential amino acid. In the same sentence “The T4 and T3, which are iodinated molecules of the essential amino acid tyrosine, regulate cellular oxidation…”, what does “cellular oxidation” mean?

Reply:

As suggested by the reviewer, I have amended the text and replaced cellular oxidation with “regulate oxidative enzymes” and amended the sentence regarding tyrosine. While it is general accepted that tyrosine is a non-essential amino acid, I had included wording of “essential” based on the finding of Bross et al  Am J Physiol Endocrinol Metab. 2000 Feb;278(2):E195-201. The sentence has been revised to read as follows:

“The T4 and T3, which are iodinated molecules of tyrosine, an amino acid that is synthesized in the body from phenylalanine, that is an essential amino acid, regulate oxidative enzymes and hence affect calorigenesis, thermoregulation, and intermediary metabolism. In addition to regulating carbohydrate metabolism these hormones also stimulate protein synthesis”.

2)      First paragraph, line 38: “These hormones are necessary for protein synthesis”. Thyroid hormones stimulate the synthesis of some proteins, but this sentence seems to be very generic and should be rewritten.

Reply: As suggested by reviewer, I have amended this sentence.

3)      Page 3, 132: “update” might be changed to “uptake”.

Reply: The correction has been made.

4)      Page 4, Metabolism of Iodine, second paragraph, line 177: The sentence “The colloid is a gycoprotein called thyroglobulin” should be changed to “The colloid contains a gycoprotein called thyroglobulin”.

Reply: The correction has been made.

5)      Same paragraph, lines 183-184: The sentence “Moreover, NIS regulates the iodination of Tg leading to T4 and T3 synthesis” might be changed to “Moreover, NIS regulates iodine availability in the thyroid, necessary for Tg iodination and T4 and T3 synthesis”.

Reply; As suggested by the reviewer the sentence has been changed.

6)      Same paragraph, lines 189-190: “To minimize the deleterious effects of iodide deficiency, iodinated Tg stored extracellularly [79] and this process is mediated by megalin [80-82].” Megalin is important for Tg transcytosis. Is it involved in Tg storage too?

Reply: The text has been revised as suggested to provide a clearer definition of the role of megalin in regulating Tg function.

7)      Page 5, second paragraph: “During lactation, the mammary gland concentrates iodine and secretes it into breast milk to provide for the new-born, until it can synthesize its own TH [96].” Thyroid hormones are synthetized by the fetus early in gestation. The sentence does not make sense.

Reply: The text has been revised as suggested.

8)      Page 7, last paragraph, line 342: The sentence “…NIS uptake of iodine into Tg.” does not make sense and should be rewritten.

Reply: The text has been revised as suggested.

9)      Page 10, first paragraph, line 455-456: The sentence “Taken together, this suggests that F intake be minimized in infancy…” might be changed to “Taken together, this suggests that F intake should be minimized in infancy…”

Reply: The text has been revised as suggested.

10)  Page 11, conclusions, line 513: “…iodine update…” should be changed to “…iodide uptake…”.

Reply: The text has been revised as suggested.

Reviewer 2 Report

This is an interesting useful review paper that focuses on  the effect of fluoride exposure on thyroid function which has not been reviewed thoroughly in the past. The references to the bibliography are extensive andthe review is written in a way that readers from both basic research and clinical background can follow and understand. 

This review does not only focus on the studies that show a direct effect of fluoride on thyroid through the Na/K ATPase inhibition but also reviews studies that show the potential of other pathways to inhibit NIS and analyzes the potential effects of fluoride on these other pathways. 

The minor remarks that I have to do are the following:

a. The pathways of Tg, TNF-α,TGF-β1,IL6, IL-1β,IFN-γ,IGF-1and  PI3K have known roles to modulate NIS activity but the way fluoride affects them is not well established. In each instance the authro describes an effect of fluoride on this pathway it should be stated if in the same study the effect on NIS and on thyroid physiology has actually been evaluated or it was just a speculation.  

b. Moreover, a simple model figure that summarizes the effects of fluoride on NIS directly or indirectly through Tg, TNF-a etc would be a good addition to this manuscript as a visual summary/take home message of the review.

c. Minor typos should be double checked such as:

259: precious--> previous

Author Response

I would like to thank the reviewer for careful and thorough reading of this manuscript and for the thoughtful comments and constructive suggestions, which help to improve the quality of this manuscript.

General Comments. This is an interesting useful review paper that focuses on  the effect of fluoride exposure on thyroid function which has not been reviewed thoroughly in the past. The references to the bibliography are extensive and the review is written in a way that readers from both basic research and clinical background can follow and understand. This review does not only focus on the studies that show a direct effect of fluoride on thyroid through the Na/K ATPase inhibition but also reviews studies that show the potential of other pathways to inhibit NIS and analyzes the potential effects of fluoride on these other pathways.

Reply:

I appreciate the positive feedback from the reviewer.

Minor comments:

1)      The pathways of Tg, TNF-α,TGF-β1,IL6, IL-1β,IFN-γ,IGF-1and  PI3K have known roles to modulate NIS activity but the way fluoride affects them is not well established. In each instance the author describes an effect of fluoride on this pathway it should be stated if in the same study the effect on NIS and on thyroid physiology has actually been evaluated or it was just a speculation. 

Reply:

As suggested by the reviewer, I have amended the text accordingly. The changes are highlighted in section 2.2 lines 272-275. It is important to note that there are no studies in published literature addressing the molecular mechanisms of fluoride inhibition of NIS activity. This study has therefore identified from published literature the known and established mechanisms/pathways of inhibition of NIS expression or activity and sought evidence from published literature of the effects of fluoride on these biological pathways.

2)      b. Moreover, a simple model figure that summarizes the effects of fluoride on NIS directly or indirectly through Tg, TNF-a etc would be a good addition to this manuscript as a visual summary/take home message of the review.

Reply. I have included Figure 1, a schematic illustration of the role of NIS and Na+, K+-ATPase in normal thyroid follicular cells showing the key aspects of  iodine transport, thyroid hormone synthesis and molecular mechanisms of inhibition of NIS by fluoride.

Int. J. Environ. Res. Public Health EISSN 1660-4601 Published by MDPI AG, Basel, Switzerland RSS E-Mail Table of Contents Alert
Back to Top